# Sort Before You Prune: Improved Worst-Case Guarantees of the DiskANN Family of Graphs

Siddharth Gollapudi [1]  Ravishankar Krishnaswamy [2]  Kirankumar Shiragur [2]  Harsh Wardhan [2]

## Abstract

Graph-based data structures have become powerful and ubiquitous tools for scalable approximate nearest-neighbor (ANN) search over the past decade. In spite of their apparent practical performance, there has only recently been progress on the *worst-case* performance of these data structures. Indeed, the influential work of Indyk & Xu introduced the key concept of $\alpha$-reachable graphs, showing that graphs constructed by the DiskANN algorithm (Subramanya et al., 2019) produce an $\left(\frac{\alpha+1}{\alpha-1}\right)$-approximate solution with a simple best-first search that runs in poly-logarithmic query time. In our work, we improve and generalize this analysis as follows:

- We introduce *sorted* $\alpha$-reachable graphs, and use this notion to obtain a stronger approximation factor of $\frac{\alpha}{\alpha-1}$ for the DiskANN algorithm on Euclidean metrics.
- We present the *first* worst-case theoretical analysis for the popular *beam-search* algorithm, which is used in practice to search these graphs for $k > 1$ candidate nearest neighbors.

We also present empirical results validating the significance of sorted $\alpha$-reachable graphs, which aligns with our theoretical findings.

## 1. Introduction

In the nearest-neighbor search (NNS) problem, there is an underlying metric space $(X, D)$, where $X$ is a set of points and $D : X \times X \to \mathbb{R}_{\geq 0}$ is a distance function between points in $X$. Given a dataset $P \subseteq X$ of $n$ points, the goal is to design an efficient data structure which, for any query point $q \in X$, and target $k$, efficiently identifies the $k$ nearest neighbors of $q$ from $P$. In real-world applications, these points can be thought of as mathematical representations of online data, such as documents, images, or user behaviors, with their distances being an appropriately chosen metric, such as Euclidean or cosine similarity for vector representations of objects, or Jaccard similarity when the dataset $P$ consists of sets of words. In these settings, the approximate version of the problem, abbreviated ANNS, is usually considered instead. For this problem, the data structure is allowed to return sub-optimal results, and its quality is typically measured by the approximation ratio: the ratio of the distances of the $k^{\text{th}}$ furthest point identified by the algorithm to that of the true $k^{\text{th}}$ furthest point in the dataset. In more applied papers, a notion of *recall@k* is instead used to measure the quality of a solution, which is the fraction of true $k$ nearest neighbors of a query that are computed by the algorithm in its $k$ candidate near neighbors, averaged over all queries.

The ANNS problem has been extensively researched in the theoretical as well as applied communities (Beygelzimer et al., 2006; Babenko & Lempitsky, 2014; Johnson et al., 2017; Weber et al., 1998; Baranchuk et al., 2018; Malkov & Yashunin, 2016; Jégou et al., 2011; Arya & Mount, 1993; Indyk & Motwani, 1998a) and serves as the cornerstone for numerous applications across diverse domains: computer vision (Wang et al., 2012), data mining (Camerra et al., 2010), information retrieval (Manning et al., 2008), classification (Fix & Hodges, 1989), and recommendation systems (Dahiya et al., 2021), to name a few. With the recent explosion of semantic search powered by deep learning models (Devlin et al., 2018) and large language models, the problem is playing an even larger role in powering the AI revolution.

Initial ANNS results have focused on smaller datasets (on the order of millions of points represented as vectors in tens of dimensions), developing a plethora of breakthrough techniques centered around space partitioning (Johnson et al., 2017), such as Locality Sensitive Hashing (Indyk & Motwani, 1998b; Andoni & Indyk, 2008), KD-trees (Arya et al., 1998), and cover trees (Beygelzimer et al., 2006), to name a

---

*Authors listed alphabetically. [1]UC Berkeley [2]Microsoft Research India. Correspondence to: Siddharth Gollapudi <sgollapu@berkeley.edu>.

*Proceedings of the 42$^{nd}$ International Conference on Machine Learning*, Vancouver, Canada. PMLR 267, 2025. Copyright 2025 by the author(s).

few. A potential drawback of these approaches is that the data structure typically needs to scan a very large (exponential in the dimension) number of neighboring partitions around a given query's partition to retrieve candidate nearest neighbors, and the problem of quickly identifying these nearby cells to scan becomes challenging. To circumvent these so-called "boundary effects," there has been an evolution of *graph-based ANNS indexing algorithms* (Malkov & Yashunin, 2016; Fu et al., 2019; Iwasaki & Miyazaki, 2018; Sugawara et al., 2016; Iwasaki; Subramanya et al., 2019), which construct a navigation graph over the dataset $P$, allowing for a simple greedy walk search algorithm on the graph, which keeps on traversing neighbors provided they are closer to the query w.r.t the metric $D$. Many different comparative studies (Aumüller et al., 2020; Echihabi et al., 2019; Li et al., 2020; Wang et al., 2021) of ANNS algorithms have concluded that these graph-based methods significantly outperform other techniques in terms of search performance on a range of real-world static datasets, and they enjoy usage in industry at scale (Simhadri et al., 2023).

Given the strong empirical performance of these graph-based algorithms, it is crucial to understand how they work and to establish provable guarantees on their behavior. Recent studies have analyzed the properties of the $k$-nearest neighbor graph for ANNS—where each datapoint connects to its $k$ closest neighbors, and search proceeds via a greedy walk (Laarhoven, 2017)—as well as the impact of adding long-range edges in $k$-nearest neighbor graphs (Prokhorenkova & Shekhovtsov, 2020) for *random datasets*, where points are uniformly sampled from the unit sphere in $d$ dimensions, and the query is placed near a datapoint.

More broadly, Indyk & Xu introduce a class of graphs known as $\alpha$-reachable graphs (the definition of which is motivated by the DiskANN algorithm), achieving *provable worst-case space and time complexity bounds for any dataset*. The importance of this work comes from the fact that it sets apart the structural properties inherent in the DiskANN graph construction from the other methods such as HNSW and NSG, which have $\alpha$ set implicitly to one and thus come without provable worst-case guarantees. Intuitively, having $\alpha > 1$ ensures that there are sufficient long-range edges present when the graph is constructed. While it takes an important step towards a theoretical understanding of graph-based ANNS algorithms, the work left three important questions, which form the crux of this work:

> 1. Indyk & Xu proved that the DiskANN graph construction algorithm achieves an approximation factor of $\frac{\alpha+1}{\alpha-1}$ for *any* metric. Can we leverage the fact that most modern applications use the Euclidean $\ell_2$ metric to derive stronger guarantees?

> 2. The original DiskANN algorithm constructs edges for each node by applying a pruning procedure (with a parameter $\alpha > 1$) to a sorted list of potential candidate edges. However, Indyk and Xu (Indyk & Xu, 2024)'s proof does not explicitly make use of this sortedness of the list to obtain their guarantees. Is this then needed in the original DiskANN algorithm, or can we leverage it to improve our theoretical bounds?

> 3. The proof in (Indyk & Xu, 2024) provides guarantees only for the single best neighbor found via greedy search, relative to the true nearest neighbor of the query. However, in practice, we often retrieve $k > 1$ candidates using a generalized greedy search known as BeamSearch. Can we obtain any provable guarantees for beam search to get $k > 1$ results?

**Our Contributions** In this paper, we address all three of these questions. Most interestingly, we show that questions one and two alone cannot offer any improvements, but when combined, can be used to obtain improved analysis of the DiskANN algorithm! Intuitively, it establishes that the sorting step before pruning candidates offers theoretical benefits, and we validate this empirically. Next, we show that our improved analysis techniques can also be used to derive approximation guarantees when using beam search to compute $k$ candidate results. Crucially, to the best of our knowledge, this is the first such result which obtains provable worst-case guarantees for graph methods when requiring to identify $k > 1$ candidate nearest neighbors, which is one of the most common uses of this problem in practice:

> **Theorem 1.1** (Final convergence bounds). *Let $G$ be a DiskANN graph. For any query $q$,* BeamSearch *of size $L$ in at most $O(L + \log_\alpha \frac{\delta}{(\alpha-1)\epsilon})$ steps outputs a set of points $\{b_1, \ldots, b_L\}$ such that each $b_j$ satisfies:*
>
> $$D(b_j, q) \le \epsilon + \frac{\alpha}{\alpha - 1} \cdot D(a_j, q) \text{ for } \ell_2 \text{ metric } D$$
> $$D(b_j, q) \le \epsilon + \frac{\alpha + 1}{\alpha - 1} \cdot D(a_j, q) \text{ for any metric } D$$
>
> *where $a_j$ is the $j$th nearest neighbor to $q$.*

Furthermore, the number of distance comparisons per step in the BeamSearch algorithm is upper bounded by the maximum degree of the DiskANN graph, which Indyk & Xu showed to be $O\left((4\alpha)^\Delta \log \delta\right)$, where $\delta$ and $\Delta$ denote the dataset's maximum distance and doubling dimension, respectively. See Section 2 for definitions and further details.

# 2. Preliminaries

**Notation.** Consider a dataset $P$ of $n$ points in a metric space $(X, D)$. For any query point $q \in X$, a top-$k$ nearest-neighbor search ($k$-NNS) data structure returns the $k$ closest points to $q$ in $P$ based on the distance function $D$. Since exact nearest-neighbor search suffers from the curse of dimensionality (Clarkson, 1994), approximate methods are often preferred. In particular, under the following approximation model, if $p \in P$ is the $i$th nearest neighbor of $q$ ($i \leq k$), a point $p' \in P$ is considered a $(1 + \epsilon)$-approximation to $p$ if $D(p', q) \leq (1 + \epsilon) \cdot D(p, q)$.

All graphs considered will be directed and represented as $G = (V, E)$, where $V$ is the set of nodes and $E$ is the set of directed edges of the form $(u, v)$ or $u \rightarrow v$. For any node $u \in V$, let $N_{\text{out}}(u)$ denote its set of out-neighbors, i.e., $\{v \mid (u, v) \in E\}$. In graph-based ANNS index structures, the graph nodes correspond one-to-one with the dataset points in $P$. To simplify notation, we use $P$ to refer to both the dataset and the graph nodes. Thus, an edge $(p, p')$ in the graph represents a directed link from $p$ to $p'$, while $D(p, p')$ denotes the metric distance between the corresponding dataset points. Similarly, $D(p, q)$ represents the distance from $p$ to the query point $q$.

## 2.1. DiskANN Algorithm Overview

Here we provide a brief overview of the key components of the DiskANN data structure and refer the reader to the original work (Subramanya et al., 2019) for further details. At a high level, given a dataset $P$, DiskANN constructs a graph with $|P|$ nodes, designed so that a simple greedy search algorithm efficiently returns good approximate nearest neighbors for a query $q$. The greedy search starts from a dedicated start node $s$ and iteratively moves to the neighboring node that is closest to $q$ based on the metric $D$, stopping when all neighbors are farther than the current node.

A generalized version of this search, described in Algorithm 1, maintains a priority queue of size $L \geq 1$, continuously refining the queue until a local optimum is reached. This approach, known as BeamSearch, is the standard search strategy for graph-based ANN algorithms (HNSW; Subramanya et al., 2019).

The construction of a graph for the dataset $P$ can be approached in two ways, as presented in (Subramanya et al., 2019) and later formalized in (Indyk & Xu, 2024): a *slow-preprocessing* variant that illustrates the fundamental graph construction principles, and a *fast-preprocessing* variant that provides an efficient, scalable heuristic approximation for large datasets.

In both variants, each node $p$ selects neighbors from a candidate set $\mathcal{E}$ using a Prune procedure Algorithm 2, which ensures that an edge $(p, p')$ is retained only if $D(p, p') <$

---

**Algorithm 1** BeamSearch$(s, q, k, L)$

---

**Require:** Graph $G$ with start node $s$, query $q$, result size $k$, search list size $L \geq k$

**Ensure:** Result set $\mathcal{L}$ containing $k$-approx nearest neighbors, and a set $\mathcal{V}$ containing all the visited nodes

1: Initialize sets $\mathcal{L} \leftarrow \{s\}$, $\mathcal{E} \leftarrow \emptyset$, and $\mathcal{V} \leftarrow \emptyset$ {$\mathcal{L}$ is the list of best $L$ nodes, $\mathcal{E}$ is the set of all nodes which have already been expanded from the list, and $\mathcal{V}$ is the set of all nodes which have been visited, i.e., inserted into the list}
2: **while** $\mathcal{L} \setminus \mathcal{E} \neq \emptyset$ **do**
3:     Let $p^* \leftarrow \arg\min_{p \in \mathcal{L} \setminus \mathcal{E}} D(p, q)$
4:     Update $\mathcal{L} \leftarrow \mathcal{L} \cup (N_{\text{out}}(p^*) \setminus \mathcal{V})$ and $\mathcal{E} \leftarrow \mathcal{E} \cup \{p^*\}$
5:     **if** $|\mathcal{L}| > L$ **then**
6:         Update $\mathcal{L}$ to retain closest $L$ points to $q$
7:     **end if**
8:     Update $\mathcal{V} \leftarrow \mathcal{V} \cup N_{\text{out}}(p^*)$
9: **end while**
10: **return** closest $k$ points from $\mathcal{V}$; $\mathcal{V}$

---

$\alpha \cdot D(p^*, p')$ for some $p^*$ already connected to $p$. This prevents unnecessary edges while maintaining search efficiency.

---

**Algorithm 2** Prune$(p, \mathcal{E}, \alpha, R)$

---

**Require:** Graph $G$, point $p \in P$, candidate set $\mathcal{E} \subseteq P$, distance threshold $\alpha \geq 1$, degree bound $R$

**Ensure:** $G$ is modified by setting at most $R$ new out-neighbors for $p$

1: Update $\mathcal{E} \leftarrow (\mathcal{E} \cup N_{\text{out}}(p)) \setminus \{p\}$
2: Initialize $N_{\text{out}}(p) \leftarrow \emptyset$
3: **while** $\mathcal{E} \neq \emptyset$ **do**
4:     Let $p^* \leftarrow \arg\min_{p' \in \mathcal{E}} D(p, p')$
5:     Update $N_{\text{out}}(p) \leftarrow N_{\text{out}}(p) \cup \{p^*\}$
6:     **if** $|N_{\text{out}}(p)| = R$ **then**
7:         break
8:     **end if**
9:     **for** $p' \in \mathcal{E}$ **do**
10:         **if** $\alpha \cdot D(p^*, p') \leq D(p, p')$ **then**
11:             Update $\mathcal{E} \leftarrow \mathcal{E} \setminus \{p'\}$
12:         **end if**
13:     **end for**
14: **end while**

---

The slow-preprocessing variant sets $\mathcal{E} = P \setminus \{p\}$ and allows a maximum degree of $R = n - 1$, leading to a $O(n^3)$ runtime, which is impractical for large datasets. However, the resulting graph is sparse when the dataset has low intrinsic dimension (formalized in Section 2.2), and the greedy search converges to a good approximate solution (Indyk & Xu, 2024). The fast-preprocessing variant mitigates the high computational cost by capping the degree at a predefined parameter $R$ and restricting $\mathcal{E}$ to nodes visited during the greedy search in Algorithm 1 before finalizing $p$'s edges. This significantly accelerates graph construction while preserving search quality.

## 2.2. Overview of Indyk-Xu (Indyk & Xu, 2024)

For convenience, we summarize the key results from Indyk & Xu, as our work builds upon and significantly improves them. We start with the definition of $\alpha$-reachable graphs.

**Definition 2.1.** A directed graph $G$ is said to be $\alpha$-reachable for $\alpha \geq 1$ if, for any two nodes $v, a \in G$, either the edge $(v, a)$ exists or there exists a node $t \in G$ such that the edge $(v, t)$ exists *and* $D(t, a) \leq D(v, a)/\alpha$.

In the slow-preprocessing variant of DiskANN, every node $p$ considers all other nodes as candidate out-neighbors during the pruning procedure, with Algorithm 2 selecting the final subset to retain. Effectively, this means $p$'s out-neighbors are given by $\text{Prune}(p, P \setminus \{p\}, \alpha, n-1)$. For such a graph, Indyk and Xu established the following results.

**Lemma 2.2** ($\alpha$-reachable lemma). *For every node $p \in P$, if $p$ is connected to the output of $\text{Prune}(p, P \setminus \{p\}, \alpha, n-1)$, then $G(P, E)$ is $\alpha$-reachable.*

This $\alpha$-reachability property is crucial for ensuring that the greedy search algorithm converges to high-quality solutions. Before demonstrating this, we first establish that the $\text{Prune}$ algorithm produces *sparse* graphs, which in turn allows us to bound the search complexity of the greedy algorithm.

The sparsity of these graphs is governed by the intrinsic dimension of the data set, quantified by *doubling dimension*. A dataset $P$ has a doubling constant $C$ if, for any point $p \in P$ and radius $r > 0$, the set $P \cap B(p, 2r)$ can be covered by at most $C$ balls of radius $r$. The *doubling dimension* $\Delta$ is then defined as $\log C$. In this work, we assume that the datasets have a bounded doubling dimension, a reasonable assumption since many real-world datasets exhibit a significantly lower intrinsic dimensionality than their ambient space (Aumüller & Ceccarello, 2019; Cayton et al., 2008).

**Lemma 2.3** (Degree bound lemma). *For every node $p \in P$, if $p$ is connected to the output of $\text{Prune}(p, P \setminus \{p\}, \alpha, n-1)$, then the degree of $p$ is at most $O\left((4\alpha)^\Delta \log \delta\right)$, where $\Delta$ and $\delta$ denote the doubling dimension and the aspect ratio of $P$.*

Indyk and Xu rely on these results to establish their main convergence theorem, stated next.

**Theorem 2.4** (Convergence theorem). *Let $G(P, E)$ be a DiskANN graph constructed with the "slow-preprocessing" variant as in Lemma 2.2. Then, for any starting point $s \in P$ and beam-size $L = 1$, $\text{BeamSearch}(s, q, 1, 1)$ outputs an $\left(\frac{\alpha+1}{\alpha-1} + \epsilon\right)$-approximate nearest neighbor in $O\left(\log_\alpha \frac{\delta}{(\alpha-1)\epsilon}\right)$ steps and $O\left((4\alpha)^\Delta \log \delta\right)$ distance comparisons per step.*

Note that Theorem 2.4 applies only when the beam size is 1. In practice, a larger beam size is typically used, leading to

improved empirical performance and enabling the retrieval of multiple ($k > 1$) nearest neighbors.

## 3. Overview of Proof

Here, we provide an overview of the proof for our main result, which offers an improved analysis of the DiskANN algorithm and extends it to the beam search setting. Our contributions follow a natural sequence: we first identify a stronger structural property of DiskANN graphs, use this to derive tighter approximation bounds (especially in Euclidean space), and ultimately provide the first provable guarantees for beam search in graph-based data structures. The key insight behind our analysis is that DiskANN's graph construction implicitly maintains a "sorting" property that, when properly leveraged, enables significantly stronger theoretical guarantees.

The starting point of our analysis is a subtle yet powerful observation about DiskANN's behavior. Specifically, consider how DiskANN determines the out-neighbors of a node $p$ from a set of candidates $\mathcal{E}$ using Algorithm 2. While the previous analysis in Lemma 2.2, which shows $\alpha$-reachability, would hold even without step (4) of $\text{Prune}$ — meaning we could select any node $p^*$ from $\mathcal{E}$ — DiskANN explicitly chooses the closest remaining node to $p$. This seemingly minor implementation detail has significant theoretical implications for the quality of the computed solutions. We now formalize this insight, with a stronger notion of reachability.

**Definition 3.1** (Sorted $\alpha$-reachable graphs). Given a dataset $P$, a distance metric $D$ and $\alpha > 1$, we call a directed graph $G$ to be a *sorted* $\alpha$-reachable graph if for any pair of points $v, a \in P$, either the edge $(v, a)$ exists in $G$ or there exists a point $t \in P$ such that:

1. $(v, t)$ is an edge in $G$

2. $D(t, a) \leq D(v, a)/\alpha$ ($\alpha$-reachability)

3. $D(v, t) \leq D(v, a)$ (sorting property)

The first two properties define standard $\alpha$-reachability. The third property, which we refer to as the sorting property, ensures that each vertex connects to closer points before farther ones. This property naturally emerges from DiskANN's graph construction and, as we will demonstrate, plays a crucial role in enabling a tighter analysis in Euclidean space. In this context, Lemma 2.2, combined with the sorting step implicit in Algorithm 2, leads to the following result.

**Lemma 3.2** (Sorted $\alpha$-reachable lemma). *Given a dataset $P$, a distance metric $D$ and an $\alpha > 1$, the "slow preprocessing" variant of DiskANN algorithm constructs a sorted $\alpha$-reachable graph with respect to distance metric $D$.*

We defer the formal proof of the above result to Appendix A.1. Given that DiskANN constructs this specialized

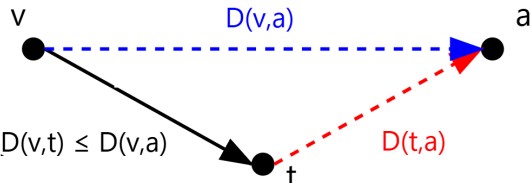

Figure 1: Sorted $\alpha$-reachablilty property. The intermediate node $t$ must satisfy both $D(v,t) \leq D(v,a)$ and $D(t,a) \leq D(v,a)/\alpha$.

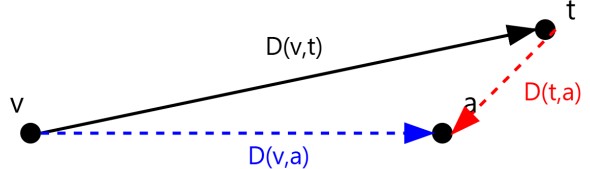

Figure 2: Vanilla $\alpha$-reachability property. The intermediate node $t$ only needs to satisfy $D(t,a) \leq D(v,a)/\alpha$.

class of sorted $\alpha$-reachable graphs, we now analyze the quality of solutions returned by beam search on these graphs. Indyk and Xu (Indyk & Xu, 2024) conducted a similar analysis for $\alpha$-reachable graphs, deriving a recurrence relation that shows how, at each step of the greedy search algorithm, either the distance to the query decreases geometrically or the search is already sufficiently close to the query. They then use a case analysis to establish the overall approximation quality upon termination. For clarity, we first analyze the local optimality of solutions returned by the beam search algorithm on sorted $\alpha$-reachable graphs before presenting the convergence analysis.

### 3.1. Approximation Quality of Local Optimum using Beam Search

We begin by analyzing the quality of a simple greedy search, that is, where the beam size is 1. We take a different approach from (Indyk & Xu, 2024) in our analysis, to fully exploit the structure of the problem. To this end, suppose that we have a query $q$, for which the greedy search *has already terminated* to a local optimum point $v$, and suppose the true nearest neighbor of $q$ is a data point $a$. What can the worst possible termination be, for this algorithm? Indeed, it cannot be an arbitrary arrangement of points $q$, $v$, and $a$, since the algorithm has some structural properties. For example, it must be the case that $v$ *does not have an edge* to $a$, as otherwise a greedy search would have walked along the edge $(v, a)$ to discover the global optimum $a$. But when can this edge be pruned? This happens when there exists another point $t$ which is responsible for $(v, a)$ being pruned, which satisfies additional conditions as per Definition 3.1. The crucial observation then is that the approximation ratio achieved by any local optimum solution $v$ can in fact be *upper bounded by the following optimization problem.*

$$\alpha_{\text{opt}} = \max D(t,q) \quad \text{such that:} \qquad (1)$$
$$D(q,v) \leq D(q,t), \ D(a,t) \leq D(v,a)/\alpha$$
$$D(v,t) \leq D(v,a), \ D(a,q) = 1$$
$$D(\cdot,\cdot) \text{ is a metric, i.e., satisfies triangle inequalities}$$

In the above optimization problem we seek to find the worst possible arrangement of points which satisfies the structural requirements of our algorithm. The first constraint $D(q,v) \leq D(q,t)$ captures the fact that our current solution $v$ is locally optimal, the constraint $D(a,t) \leq D(v,a)/\alpha$ follows from the $\alpha$-reachability property of the graph, the constraint $D(v,t) \leq D(v,a)$ follows from the sorting property, and finally, $D(a,q) = 1$ is a standard normalization step for analyzing ratios. Indeed, the approximation ratio is actually $D(v,q)/D(a,q)$ which we upper bound by maximizing $D(t,q)$ (note that $D(v,q) \leq D(t,q)$ and so this is a valid relaxation) while enforcing $D(a,q) = 1$. The part where we replaced $D(v,q)$ by $D(t,q)$ in the objective function will help in analyzing the beam search result as well.

In our first result we upper bound the objective value of the above optimization problem for the general metric.

**Lemma 3.3** (Objective value for general metric). *When $D$ is a general metric, the optimum value of optimization problem 1 is upper bounded by $\frac{\alpha+1}{\alpha-1}$.*

The proof of this result follows from the triangle inequality and the $\alpha$-reachable property; see Appendix A.2 for a formal proof. This result provides an alternative formulation of the findings in (Indyk & Xu, 2024). Next, we strengthen this guarantee for the important case of *Euclidean metrics*. The proof can be found in Appendix A.3.

**Lemma 3.4** (Objective value for Euclidean metric). *When $D$ is the Euclidean $\ell_2$ metric, the optimum value of optimization problem 1 is upper bounded by $\frac{\alpha}{\alpha-1}$.*

Intuitively, the strengthening comes because for general metrics, the optimization problem ends up being a linear program over the sixteen variables of the form $D(x,y)$ where $x, y \in \{v, q, t, a\}$, which we can optimize. However, when the metric is Euclidean, we can re-write the optimization problem in terms of dot products (by considering squared distances in the first four constraints as well as the objective function), and replace the metric constraint with the much stronger constraint that enforces that the matrix of the dot-product variables is positive-semidefinite. We can then

solve the resulting SDP using a solver to get the desired bounds for any given $\alpha$. In this paper, we actually give a formal geometric proof for any $\alpha$ without resorting to the use of solvers, by using SDP duality. The above two results provide an immediate bound on the quality of the local optimum solution, which we summarize below.

**Corollary 3.5** (Guarantee of greedy search). *For any dataset $P$, and query $q$, the greedy search algorithm terminates at a local optimum point $v$ which is an $\frac{\alpha+1}{\alpha-1}$-approximation to the true nearest neighbor of $q$ for arbitrary metrics, which improves to $\frac{\alpha}{\alpha-1}$ for Euclidean metrics.*

While the previous result establishes bounds for the nearest neighbor candidate returned by the algorithm, we now extend this analysis to beam search, where we maintain a beam $B_L = \{b_1, \ldots, b_L\}$ of $L$ candidates.

To analyze beam search, our approach is the following: let $a$ denote the closest member in the optimal solution of $L$ nearest neighbors which is *not present* in the locally optimal beam, and let $v$ be the element in the locally optimum beam which is *closest to* $a$. Clearly, $v$ does not have an edge to $a$, since otherwise, the beam would improve by including $a$ and evicting $b_L$. Therefore, we can define $t$ to be the node responsible for blocking the edge $(v, a)$, based on the sorted $\alpha$-reachable condition between $v$ and $a$. Moreover, since $t$ is closer to $a$ than $v$, it *cannot be part of the beam!* by virtue of the choice of $v$. However, because the beam has an outgoing edge to $t$, we get $D(t, q)$ as an upper bound on $D(b_i, q)$ for all the elements $b_i \in B_L$ in the beam, *in one shot*! We can thus reuse optimization problem 1 to derive an upper bound on the quality of all points in the beam.

**Lemma 3.6** (Guarantee of beam search). *Let $G$ be a sorted $\alpha$-reachable graph with respect to distance function $D$. For any query point $q \in \mathbb{R}^d$, let $B_L = \{b_1, \ldots, b_L\}$ be a local optimum solution of size $L$ with respect to the* BeamSearch *algorithm on $G$. For each $j \in [L]$, define $B_j = \{b_1, \ldots, b_j\}$ as the first $j$ points in the beam. Then each $b_j$ satisfies:*

$$D(b_j, q) \leq \frac{\alpha}{\alpha - 1} \underset{p \in G, p \notin B_{j-1}}{\arg\min} D(q, p) \text{ for } \ell_2 \text{ metric } D$$

$$D(b_j, q) \leq \frac{\alpha + 1}{\alpha - 1} \underset{p \in G, p \notin B_{j-1}}{\arg\min} D(q, p) \text{ for general metric } D$$

This pointwise guarantee immediately demonstrates that the beam search maintains quality approximations for each position in the beam as summarized below.

**Theorem 3.7** (Pointwise guarantee of beam search). *Let $G$ be a sorted $\alpha$-reachable graph with respect to distance function $D$. For any query point $q \in \mathbb{R}^d$, let $B_L = \{b_1, \ldots, b_L\}$ be a local optimum solution of size $L$ returned by the* BeamSearch *algorithm on $G$. Then each $b_j$ satisfies:*

$$D(b_j, q) \leq \frac{\alpha}{\alpha - 1} \cdot D(a_j, q) \text{ for } \ell_2 \text{ metric } D$$

$$D(b_j, q) \leq \frac{\alpha + 1}{\alpha - 1} \cdot D(a_j, q) \text{ for general metric } D$$

*where $a_j$ is the $j^{\text{th}}$ nearest neighbor to $q$.*

The improvement from $\frac{\alpha+1}{\alpha-1}$ to $\frac{\alpha}{\alpha-1}$ in the Euclidean metric follows directly from the sorting property $D(v, t) \leq D(v, a)$ and is tight. In Appendix A.5, we provide beam search examples that confirm the tightness of our bounds.

Additionally, we show that without the sorting constraint, the $\frac{\alpha+1}{\alpha-1}$ bound remains tight even in Euclidean space. In Appendix A.6, we construct an example illustrating this, highlighting the critical role of the sorting property.

## 3.2. Convergence Rates of Beam Search

In the previous section, we analyzed the local optimum solution returned by beam search for an arbitrary number of candidates $L$. We now establish the *convergence rate*, i.e., the number of steps required for beam search to reach a good approximate solution. To do this, we modify optimization problem 1, which previously captured the guarantees of a local optimum, to instead model the progress made in each step when the current point is $\beta$ distant from the local optimum.

$$\max D(t, q) \quad \text{such that :} \tag{2}$$
$$D(a, t) \leq \frac{D(v, a)}{\alpha}, \quad D(v, t) \leq D(v, a),$$
$$D(a, q) = 1, \text{ and } D(q, v) \leq \beta + \alpha_{\text{opt}},$$
$$D(\cdot, \cdot) \text{ is a metric, i.e., satisfies triangle inequalities}$$

In comparison to optimization problem 1, the key difference is the addition of the constraint $D(q, v) \leq \alpha_{\text{opt}} + \beta$, which bounds the distance from the local optimal solution. Here, $\beta$ quantifies the deviation from local optimality, and it should decrease with each step. To establish this, we bound the objective value of the new optimization problem and show that the point $t$, identified via the sorted $\alpha$-reachable property as a neighbor of $v$, satisfies $D(t, q) \leq \alpha_{\text{opt}} + \beta/\alpha$. Thus, by moving to $t$, we achieve progress by a multiplicative factor of $\alpha$.

**Lemma 3.8** (Objective value bound for general metric). *In optimization problem 2, the maximal value of $D(t, q)$ is at most $\frac{\alpha+1}{\alpha-1} + \frac{\beta}{\alpha}$ under general metric $D$.*

**Lemma 3.9** (Objective value bound for Euclidean metric). *In optimization problem 2, the maximal value of $D(t, q)$ is at most $\frac{\alpha}{\alpha-1} + \frac{\beta}{\alpha}$ under Euclidean metric $D$.*

The proofs of the above results follow similar reasoning as Lemma 3.3 and Lemma 3.4. Specifically, for the general metric case, it relies on the triangle inequality and the $\alpha$-reachable property, while for the Euclidean case, it involves formulating optimization problem 2 as an SDP and constructing a feasible dual solution with the required value. The full proofs are deferred to Appendix B.1 and Appendix B.2, respectively.

It follows directly from optimization problem 2 and the upper bound on the objective value that we achieve a progress factor of $\alpha$ in each step. Consequently, the beam search with size 1 converges to a good approximate solution in $\log_\alpha$ time steps. The proof for larger beam sizes follows a similar approach as our analysis of local optima for higher beam sizes. Applying the same reasoning, we derive the following main convergence result.

It is immediate from optimization problem 2 and the upper bound on its optimum value that we make a multiplicative factor $\alpha$ progress in each time step. Thus it is immediate that the beam search with size 1 converges to a good approximate solution in $\log_\alpha$ time steps. The proof for larger beam sizes follows a similar approach as our analysis of local optima for higher beam sizes. Applying the same reasoning, we derive the following main convergence result.

**Theorem 1.1** (Final convergence bounds). *Let $G$ be a DiskANN graph. For any query $q$,* BeamSearch *of size $L$ in at most $O(L + \log_\alpha \frac{\delta}{(\alpha-1)\epsilon})$ steps outputs a set of points $\{b_1, \ldots, b_L\}$ such that each $b_j$ satisfies:*

$$D(b_j, q) \leq \epsilon + \frac{\alpha}{\alpha - 1} \cdot D(a_j, q) \text{ for } \ell_2 \text{ metric } D$$

$$D(b_j, q) \leq \epsilon + \frac{\alpha + 1}{\alpha - 1} \cdot D(a_j, q) \text{ for any metric } D$$

*where $a_j$ is the $j$th nearest neighbor to $q$.*

The detailed proofs of the above convergence result is provided in the Appendix B.3. Together, these results establish that sorted $\alpha$-reachable graphs not only provide better approximation guarantees but also ensure rapid convergence to these improved solutions. Our improved bounds have significant practical implications, particularly for the common case of Euclidean distance. The reduction from $(\alpha+1)/(\alpha\text{-}1)$ to $\alpha/(\alpha\text{-}1)$ means that for typical values of $\alpha$ (e.g., $\alpha$=2), we reduce the worst-case approximation ratio from 3 to 2. This tighter theoretical guarantee helps explain DiskANN's strong empirical performance and suggests that implementations should explicitly maintain the sorting property during graph construction.

## 4. Experiments

To complement our theoretical analysis for the slow-preprocessing variant of DiskANN, we empirically study the impact of *sorting* for fast-preprocessing version of DiskANN. Specifically, we compare two implementations of DiskANN: one where we sort the candidate neighbors provided to Algorithm 2, and one where we do not. We first state the setup for our evaluations and describe our datasets, and then we explain the DiskANN implementation. We then present two figures comparing 100@100*Recall* to QPS and the average number of distance comparisons performed during search.

### 4.1. Setup

**Hardware.** We conduct all experiments on a bare-metal high-performance workstation with dual Intel Xeon Gold 5218 CPUs (32 cores, 64 threads) and 256GB DDR4 RAM.

**Datasets.** We evaluate on the following datasets: the well-known SIFT1M (Jegou et al., 2010), and three modern real-world workloads comprising of OpenAI embeddings encoding abstracts scraped from Arxiv, Cohere embeddings of passages from Wikipedia articles (Cohere, 2022), and a large-scale dataset web search dataset generated using the Microsoft SPACEV model (Xu et al., 2023). We obtained these datasets from the BigANN benchmark repository (Simhadri et al., 2024) at the url https://big-ann-benchmarks.com/, and the details regarding the datasets can be found in Table 2.

| Dataset | Dimension | Document | Query |
|---------|-----------|----------|-------|
| OpenAI | 1536 | 2,321,096 | 20,000 |
| SIFT1M | 128 | 1,000,000 | 10,000 |
| Wikipedia | 788 | 35,000,000 | 5,000 |
| SPACEV | 100 | 100,000,000 | 5,000 |

Table 1: Dataset information

**DiskANN Configuration.** We use the ParlayANN DiskANN implementation to construct our instances and to build and search indices for these datasets (Manohar et al., 2024). We use the default DiskANN parameters for an in-memory build: $\alpha$ is set to be $1.2$ for all the index builds, with max degree set to $64$ and beam-width parameter $L_{\text{build}}$ set to $100$. During search, we vary the beam-width parameter $L_{\text{search}}$. The average degree of the graph constructions by DiskANN implementation with and without sorting are summarized below.

| Dataset | Sorted | Unsorted |
|---------|--------|----------|
| OpenAI | 53.899 | 56.924 |
| SIFT1M | 44.523 | 49.153 |
| Wikipedia | 24.660 | 30.730 |
| SPACEV | 46.155 | 51.020 |

Table 2: Average Degree per Constructed Graph

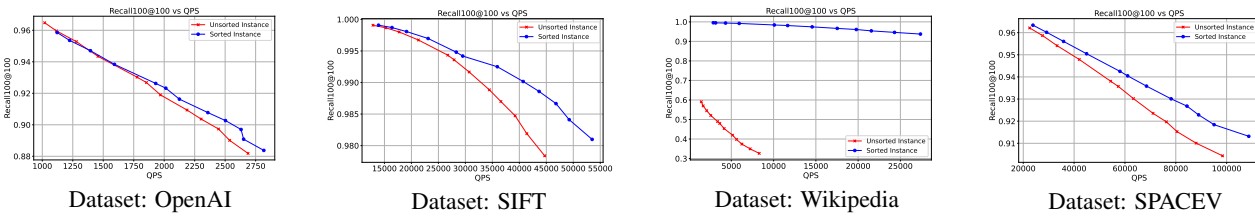

Figure 3: The above plots illustrate the Queries per Second (QPS) versus 100@100 Recall for sorted $\alpha$-reachable and standard $\alpha$-reachable graphs.

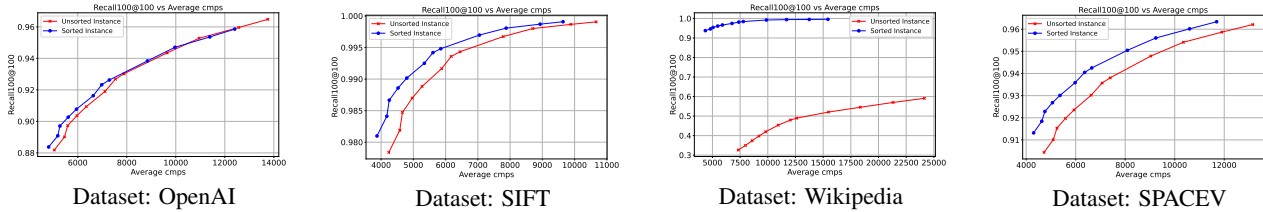

Figure 4: The above plots illustrate the average number of distance comparisons versus 100@100 Recall for sorted $\alpha$-reachable and standard $\alpha$-reachable graphs.

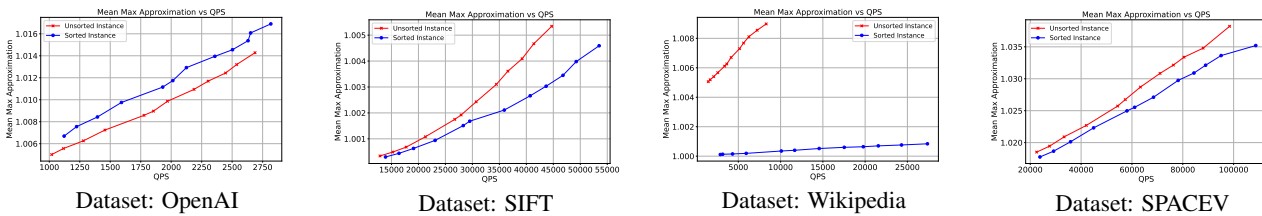

Figure 5: The above plots illustrate the average maximum approximation ratio versus 100@100 Recall for sorted $\alpha$-reachable and standard $\alpha$-reachable graphs.

| Dataset | Sorted (s) | Unsorted (s) |
|---------|-----------|--------------|
| OpenAI | 910.7 | 1049 |
| SIFT1M | 20.91 | 24.35 |
| SPACEV | 1902 | 2352 |
| Wikipedia | 6068 | 7829 |

Table 3: Build Times for Sorted and Unsorted Graphs

## 4.2. Main Empirical Evaluation

**100@100 Recall Evaluations.** For this benchmark, we evaluate the recall of our indices against a pre-computed bruteforce "groundtruth." We measure performance of these indices on basis of two metrics:

- **Queries per Second.** As mentioned, we evaluate recall along a curve by varying $L_{\text{search}}$ from 10 to 400. As $L_{\text{search}}$ increases, the corresponding *queries per second* (QPS) simply refers to the number of queries our index is able to serve per second in a *multi-threaded* setting.

- **Average Distance Comparisons.** We also measure the recall achieved at the cost of different average distance comparisons per query, which is more indicative of the qualitative impact of the sorting step, and avoids any machine specific metrics.

- **Approximation Ratio.** We verify that the approximation ratio for the sorted setting is better than the unsorted setting. We look at the average approximation ratio: the mean over all queries, of the maximum over all items $i$ in the final beam, of the distance of the $i$th candidate computed by the algorithm upon the $i$th closest NN.

## 4.3. Results

In Table 2 and Table 3, see that DiskANN consistently produces denser graphs and takes longer to build them in the unsorted setting. The unsorted index has an average degree **6%**, **10%**, **25%**, and **11%** higher than the sorted index for *OpenAI*, *SIFT1M*, *Wikipedia*, SPACEV, respectively. In addition, the unsorted index has a build time that is **13.18%**, **14.12%**, **22.48%**, and **19.13%** slower for the same datasets.

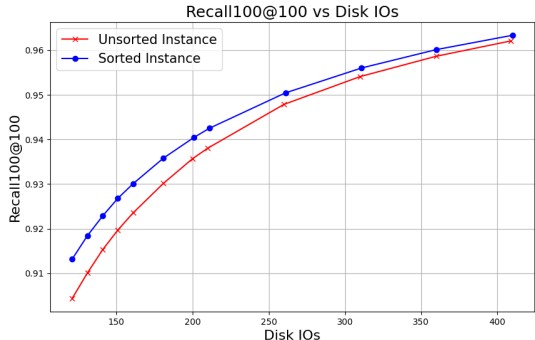

Figure 6: Plot illustrating the Disk IOs used versus the 100@100 Recall for sorted and standard $\alpha$-reachable graphs.

## 5. Conclusion

In this work, we present improved approximation guarantees of the DiskANN family of graphs for Euclidean metrics, by exploiting the sorting step the algorithm performs during index construction. We also introduce techniques to analyze the approximation quality of the widely used *beam search* algorithm for retrieving $L > 1$ candidates. Finally we experimentally validated the importance of the sorting step in index construction over diverse datasets. An interesting line of research going forward is in understanding how we can similarly exploit the structure for other real-world metrics to obtain improved guarantees, like we did for $\ell_2$ in this work.

For QPS (Figure 3), the unsorted index is slower across the board. The sorted index achieves **5%**, **8%**, and **7.5%** higher QPS for *SIFT1M* (at 99% recall), *OpenAI* (at 90% recall), and *SPACEV* (at 95% recall), respectively.

For average distance comparisons (Figure 4), the sorted index is again superior. It uses **10%** fewer comparisons for *SIFT1M* (99.9% recall), **6%** fewer for *OpenAI* (90% recall), and **18%** fewer for *SPACEV* (95% recall).

For the mean max approximation ratio, we see a difference to the results of the prior experiments: the sorted index offers superior QPS across all datasets except *OpenAI*. This is in spite of the numbers favoring the sorted indices for this dataset in the rest of our evaluation. The sorted indices are worse by **25%** on *OpenAI* (at a 1.008 approx. ratio), but better by **9%** on *SIFT* (at a 1.001 approx. ratio) and **17%** on *SPACEV* (at a 1.020 approx. ratio).

As an additional sanity check on our claims on large-scale indices, we build a disk-based index on the *SPACEV* dataset. As shown in Figure 6, the recall of the unsorted index is worse for every disk IO usage: at worst, at 400 disk IOs, the sorted index has approximately a **0.5%** improved recall.

For *Wikipedia*, the sorted index offers vastly superior performance to the unsorted index, in comparison to the smaller improvements shown for the other dataset. Along with the peculiar approximation ratio issues with the *OpenAI* dataset, we leave further study of these datasets in this setting for future work.

In conclusion, higher density graphs is usually associated with higher recall, and comes at the cost of latency. Despite the unsorted index forming denser graphs, we note that recall remains inferior to the sparser and faster sorted index, emphasizing the importance of the sorting step during index construction.

## Impact Statement

This paper presents work whose goal is to advance the field of Machine Learning. There are many potential societal consequences of our work, none which we feel must be specifically highlighted here.

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

# A. Approximation Quality of Local Optimum using Beam Search

Here we provide proof for the approximation guarantee achieved by local optimum solution of the beam search algorithm.

## A.1. Proof for Lemma 3.2

Here we show that the slow-preprocessing variant of DiskANN constructs a sorted $\alpha$-reachable graph.

**Lemma 3.2** (Sorted $\alpha$-reachable lemma)**.** *Given a dataset $P$, a distance metric $D$ and an $\alpha > 1$, the "slow preprocessing" variant of DiskANN algorithm constructs a sorted $\alpha$-reachable graph with respect to distance metric $D$.*

*Proof.* Proceed with contradiction. For an appropriate DiskANN graph $G(V, E)$, suppose there exists a pair of points $v, a \in V$ such that there does *not* exist a $t$ satisfying the conditions of Definition 3.1. We have two cases:

1. There exists an edge from $v$ to $a$. This would satisfy the requirements of Definition 3.1, and trivially contradicts our assumption.

2. There does not exist an edge from $v$ to $a$. Then, by Algorithm 2, there must exist another node $u \in V$ that was considered before $a$ that pruned $a$. This implies that $D(u, v) \leq D(a, v)$, and by the prune condition, $D(u, a) \leq D(v, a)/\alpha$. Thus, we can set $u$ to be $t$, again contradicting our assumption.

$\square$

## A.2. Proof for Lemma 3.3

Here we provide the proof for Lemma 3.3 that upper bounds the optimum value of optimization problem 1 for the case of general metric.

**Lemma 3.3** (Objective value for general metric)**.** *When $D$ is a general metric, the optimum value of optimization problem 1 is upper bounded by $\frac{\alpha+1}{\alpha-1}$.*

*Proof.* The proof of this lemma follows from a repetitive use of triangle inequality and $\alpha$-reachable property.

$$D(q, t) \leq D(q, a) + D(a, t) \leq D(q, a) + \frac{D(a, v)}{\alpha}$$

$$\leq D(q, a) + \frac{D(a, q) + D(q, v)}{\alpha} \leq D(q, a) + \frac{D(a, q) + D(q, t)}{\alpha}$$

$$\leq \frac{1}{(1 - \frac{1}{\alpha})} \cdot D(q, a)(1 + \frac{1}{\alpha}) = \frac{\alpha + 1}{\alpha - 1} \cdot D(q, a) .$$

In the first and third inequality we used triangle inequality. The second inequality uses the $\alpha$-reachability constraint. In the fourth inequality we used local optimality of $v$. Finally, in the fifth and sixth inequality, we rearranged and simplified the expression. $\square$

## A.3. Proof for Lemma 3.4

Here we provide the proof for Lemma 3.4 that upper bounds the optimum value of optimization problem 1 for the case of Euclidean metric.

**Lemma 3.4** (Objective value for Euclidean metric)**.** *When $D$ is the Euclidean $\ell_2$ metric, the optimum value of optimization problem 1 is upper bounded by $\frac{\alpha}{\alpha-1}$.*

*Proof.* We first re-state the optimization problem we wish to upper bound.

$$\max D(t, q) \quad \text{such that:} \tag{3}$$
$$D(q, v) \leq D(q, t), \ D(a, t) \leq D(v, a)/\alpha$$
$$D(v, t) \leq D(v, a), \ D(a, q) = 1$$
$$D(\cdot, \cdot) \text{ is a metric, i.e., satisfies triangle inequalities}$$

By squaring the terms, and relaxing the fourth equality as an inequality, we obtain the following *vector program*, which is essentially a semi-definite program. The goal is to find the worst configuration of vectors in euclidean space which optimize the objective.

$$\max ||q - t||^2 \quad \text{such that:} \tag{4}$$
$$||q - v||^2 \leq ||q - t||^2, \ \alpha^2||a - t||^2 \leq ||v - a||^2$$
$$||v - t||^2 \leq ||v - a||^2, \ ||q - a||^2 \leq 1$$

The optimum objective value of optimization problem 3 we wish to bound, is then the square-root of the optimum value of optimization problem 4.

See that we can leverage weak duality of semidefinite programs to upper-bound the objective function of optimization problem 4, and thus achieve an upper-bound for optimization problem 3. In order to do so, we first introduce dual variables for each of the constraints of optimization problem 4:

$$\lambda_1 \geq 0, \quad \lambda_2 \geq 0, \quad \lambda_3 \geq 0, \quad \lambda_4 \geq 0,$$

As is standard practice in deriving a dual program, we first write the Lagrangian by multiplying all constraints with their dual variables (Lagrangian multipliers) and subtracting objective function of optimization problem 4 from it.

$$\mathcal{L}(q, v, t, a) = \ ||q - t||^2 + \lambda_1 \left( ||q - v||^2 - ||q - t||^2 \right) + \lambda_2 \left( \alpha^2 ||a - t||^2 - ||v - a||^2 \right)$$
$$+ \lambda_3 \left( ||v - t||^2 - ||v - a||^2 \right) + \lambda_4 \left( ||q - a||^2 \right)$$

The resulting dual program is as follows,

$$\min \lambda_2 \text{ such that:} \tag{5}$$
$$\mathcal{L}(t, q, v, a) \geq 0$$

Now,let $\lambda_1 = \frac{1}{\alpha-1}$, $\lambda_2 = \frac{\alpha}{(2\alpha-1)(\alpha-1)}$, $\lambda_3 = \frac{\alpha}{(2\alpha-1)}$ and $\lambda_4 = \left( \frac{\alpha}{\alpha-1} \right)^2$. Note that substituting these values to dual variables, we get that, $\mathcal{L}(t, q, v, a)$ simplifies to a perfect square, thus implying non-negativity for all the values $t, q, v$, and $a$, and the feasibility of the dual constraints is satisfied. Therefore $\left( \frac{\alpha}{\alpha-1} \right)$ is a lower bound of the dual optimization function, giving an upper-bound on the optimal value for optimization problem 3 by weak duality.

$\square$

## A.4. Proof for Lemma 3.6

Here we provide proof relating the objective value of the optimization problem 1 and the approximation ratio achieved by the beam search algorithm.

**Lemma 3.6** (Guarantee of beam search)**.** *Let $G$ be a sorted $\alpha$-reachable graph with respect to distance function $D$. For any query point $q \in \mathbb{R}^d$, let $B_L = \{b_1, \ldots, b_L\}$ be a local optimum solution of size $L$ with respect to the* BeamSearch *algorithm on $G$. For each $j \in [L]$, define $B_j = \{b_1, \ldots, b_j\}$ as the first $j$ points in the beam. Then each $b_j$ satisfies:*

$$D(b_j, q) \leq \frac{\alpha}{\alpha - 1} \operatorname*{arg\,min}_{p \in G, p \notin B_{j-1}} D(q, p) \text{ for } \ell_2 \text{ metric } D$$

$$D(b_j, q) \leq \frac{\alpha + 1}{\alpha - 1} \operatorname*{arg\,min}_{p \in G, p \notin B_{j-1}} D(q, p) \text{ for general metric } D$$

*Proof.* Let $a$ be the first true nearest neighbor that is not in $B_{L-1}$. We define $v := \arg\min_{b_j \in B_L} D(b_j, a)$. Now using sorted-$\alpha$ reachability condition between $v$ and $a$ there exists a $t$ such that

$$D(t, a) \leq \frac{D(v, a)}{\alpha}$$

and $D(v, t) \leq D(v, a)$ Since current beam is locally optimal

$$D(v, q) \leq D(t, q)$$

Since $t$ is closer to $a$ than $v$, it cannot be part of the beam. However, because the beam has an outgoing edge to $t$, we get $D(t, q)$ as an upper bound on $D(b_i, q)$ for all $b_i \in B_L$. Thus we try to maximize $D(t, q)$. Now this maximization problem can be formulated as following optimization program.

$$\max \frac{D(t, q)}{D(a, q)} \text{ such that,} \qquad (6)$$

$$D(q, v) \leq D(q, t), \; D(a, t) \leq \frac{D(v, a)}{\alpha}, \; D(v, t) \leq D(v, a) \, .$$

We can always rescale this reported instance to make the distance between $a$ and $q$ a unit; hence we can reformulate optimization problem 6 as optimization problem 1.

Thus, if the optimal value of optimization problem 1 under $D$ is $\alpha_{\text{opt}}$ then

$$\frac{D(b_L, q)}{D(a, q)} \leq \frac{D(t, q)}{D(a, q)} \leq \alpha_{opt}$$

Thus after reshuffling we get

$$D(b_L, q) \leq \alpha_{\text{opt}} \arg\min_{p \in G, p \notin B_{L-1}} D(q, p)$$

Furthermore, using 3.6 and 3.4 we get the following result for the Euclidean metric,

$$D(b_L, q) \leq \frac{\alpha}{\alpha - 1} \arg\min_{p \in G, p \notin B_{L-1}} d(q, p)$$

Using 3.6 we get the following result for the general metric,

$$D(b_L, q) \leq \frac{\alpha + 1}{\alpha - 1} \arg\min_{p \in G, p \notin B_{L-1}} D(q, p)$$

$\square$

## A.5. Tight examples for local optimum solutions

**Lemma A.1.** *For any $\alpha > 1$, there exists a point set $P$ of $2L + 1$ points and a query point $q$ in 2 dimensional plane under euclidean distance function, such that there exists an locally optimal beam $\{b_1, \dots, b_L\}$ such that:*

$$d(b_j, q) = \left( \frac{\alpha}{\alpha - 1} \right) \cdot d(a_j, q)$$

*where $a_j$ is the jth nearest neighbor to q.*

*Proof.* We begin by constructing the point set $P$. Let:

- $L$ points $\{v_1, v_2, \dots, v_L\}$ be co-located at the coordinate $\left( -\frac{\alpha}{2}, 0 \right)$,

- A point $t$ be located at $\left( \frac{\alpha}{2}, 0 \right)$,

- $L$ points $\{a_1, a_2, \dots, a_L\}$ be co-located at $\left( \frac{\alpha}{2} - \frac{1}{2\alpha}, \sqrt{1 - \frac{1}{4\alpha^2}} \right)$,

- The query point $q$ be located at $\left( 0, \frac{\alpha}{2} \sqrt{\frac{2\alpha + 1}{2\alpha - 1}} \right)$.

Now, consider the sorted $\alpha$-shortcut graph over $P$. In this graph, there is no edge between any $v_i$ and $a_j$, i.e., for all $i, j$,

$$d(t, a_i) \leq \frac{d(t, v_i)}{\alpha} \quad \text{and} \quad d(t, v_i) \leq d(a_j, v_i).$$

Furthermore, we observe that:

$$d(t, q) = d(v_i, q) \quad \text{for each } i.$$

Thus, the set $\{v_1, v_2, \ldots, v_L\}$ forms a locally optimal beam.

Next, we show that the ratio $\frac{d(v_i, q)}{d(a_j, q)}$ equals $\frac{\alpha}{\alpha - 1}$.

Using the construction of the points and applying basic Euclidean distance calculations, we find:

$$\frac{d(v_i, q)}{d(a_j, q)} = \frac{\alpha}{\alpha - 1}.$$

This completes the proof, showing that the set $\{v_1, v_2, \ldots, v_L\}$ indeed forms a locally optimal beam, and the ratio of distances satisfies the given relation. $\qquad \square$

**Lemma A.2.** *For any $\alpha > 1$, there exists a point set $P$ of $2L + 1$ points and a query point $q$ in 2 dimensional plane under general metric function, such that there exists an locally optimal beam $\{b_1, \ldots, b_L\}$ such that:*

$$d(b_j, q) = \left(\frac{\alpha + 1}{\alpha - 1}\right) \cdot d(a_j, q)$$

*where $a_j$ is the jth nearest neighbor to q.*

*Proof.* We begin by constructing the point set $P$. Let:

- $L$ points $\{v_1, v_2, \ldots, v_L\}$ be colocated, such that the distance between any two points $v_i$ and $v_j$ is zero, i.e., $d(v_i, v_j) = 0$ for all $i, j$,

- A point $t$ be located such that the distance from $t$ to any $v_i$ is $d(t, v_i) = 2$,

- $L$ points $\{a_1, a_2, \ldots, a_L\}$ be colocated, such that the distance from $a_j$ to any $v_i$ is $d(a_j, v_i) = \frac{2\alpha}{\alpha - 1}$ for all $i, j$, and distance from any $a_j$ to $t$ is $\frac{2}{\alpha - 1}$,

- The query point $q$ be located such that the distance from $q$ to any $a_j$ is $d(q, a_j) = 1$, and the distance from $q$ to any $v_i$ is $d(q, v_i) = \frac{\alpha + 1}{\alpha - 1}$, and $d(q, t) = \frac{\alpha + 1}{\alpha - 1}$.

The above distance function forms a metric space. Now, consider the sorted $\alpha$-shortcut graph over $P$. In this graph, there is no edge between any $v_i$ and $a_j$, i.e., for all $i, j$,

$$d(t, a_i) \leq \frac{d(t, v_i)}{\alpha} \quad \text{and} \quad d(t, v_i) \leq d(a_j, v_i).$$

Thus, the set $\{v_1, v_2, \ldots, v_L\}$ forms a locally optimal beam. And from the distance we know the ratio $\frac{d(v_i, q)}{d(a_j, q)}$ equals $\frac{\alpha + 1}{\alpha - 1}$.

$\qquad \square$

## A.6. Tight example without sorting constraint

Here we provide an example showcasing the importance of the sorting constraint. The construction of the example is as follows. Let $t, a, q$ & $v$ all lie on x-axis at abscissa $-\frac{\alpha + 1}{\alpha - 1}, -1, 0$ and $\frac{\alpha + 1}{\alpha - 1}$ respectively. Now consider our point-set $P$ to contain only 3 points $t, a$ and $v$. A complete directed graph with edge $(v, a)$ removed is a valid $\alpha$-shortcut graph on $P$. Now for query point $q$ if the search is at $v$ it is at a locally optimal solution as $v$ and $t$ are equidistant from $q$, but approximation ratio for this case would be $\frac{D(v, q)}{D(a, q)}$ which is $\frac{\alpha + 1}{\alpha - 1}$. Note that since all these points lie on x-axis this holds for any $L_p$ norm.

# B. Convergence analysis of beam search

For convenience, we restate optimization problem 2 below.

$$\max D(t, q) \quad \text{such that:}$$
$$D(a, t) \leq \frac{D(v, a)}{\alpha}, \quad D(v, t) \leq D(v, a)$$
$$D(a, q) = 1, \quad D(q, v) \leq \beta + \alpha_{\text{opt}}$$

Recall that $\alpha_{\text{opt}}$ is optimal value of optimization problem 1 under a metric $D$.

## B.1. Proof for Lemma 3.8

Here we upper bound the objective value of optimization problem 2 for a general metric D. The formal statement and proof are given below.

**Lemma 3.8** (Objective value bound for general metric). *In optimization problem 2, the maximal value of $D(t, q)$ is at most* $\frac{\alpha+1}{\alpha-1} + \frac{\beta}{\alpha}$ *under general metric D.*

*Proof.* By triangle inequality we have that,

$$D(t, q) \leq D(t, a) + D(a, q)$$

using first constraint, that is the $\alpha$-reachable constraint, we can rewrite the above equation as,

$$D(t, q) \leq \frac{D(v, a)}{\alpha} + D(a, q)$$

Using triangle inequality on $D(v, a)$ and substituting value of $D(v, q)$ and $D(a, q)$ from the third and fourth constraint to get the desired bound as follows:

$$D(t, q) \leq \frac{D(v, q) + D(a, q)}{\alpha} + D(a, q)$$
$$\leq \frac{\beta}{\alpha} + \frac{\alpha+1}{(\alpha-1)\alpha} + \left(1 + \frac{1}{\alpha}\right) \times D(a, q) \leq \frac{\beta}{\alpha} + \frac{\alpha+1}{\alpha-1}.$$

$\square$

## B.2. Proof for Lemma 3.9

Here we upper bound the objective value of optimization problem 2 for the Euclidean metric case. The formal statement and proof are given below.

**Lemma 3.9** (Objective value bound for Euclidean metric). *In optimization problem 2, the maximal value of $D(t, q)$ is at most* $\frac{\alpha}{\alpha-1} + \frac{\beta}{\alpha}$ *under Euclidean metric D.*

*Proof.* For convenience, let D be defined by $|| \cdot ||_2^2$. In our optimization problem, we can accomplish this by squaring all terms, giving the following modified version of optimization problem 2:

$$\max D(t, q) \text{ such that:} \tag{7}$$
$$D(a, t) \leq \frac{D(v, a)}{\alpha^2}, \ D(a, q) \geq 1, \ D(a, q) \leq 1,$$
$$D(v, t) \leq D(v, a), \ D(q, v) \leq \left(\beta + \frac{\alpha}{\alpha-1}\right)^2.$$

Indeed, $|| \cdot ||_2^2$ can be expressed as dot products of vectors, allowing us to expand optimization problem 7 into the following semidefinite program:

$$\max(t^T t - 2t^T q + q^T q) \quad \text{such that:} \tag{8}$$
$$a^T a - 2a^T t + t^T t \leq \frac{v^T v - 2v^T a + a^T a}{\alpha^2}$$
$$a^T a - 2a^T q + q^T q \geq 1$$
$$a^T a - 2a^T q + q^T q \leq 1$$
$$v^T v - 2v^T t + t^T t \leq v^T v - 2v^T a + a^T a$$
$$q^T q - 2q^T v + v^T v \leq \left(\beta + \frac{\alpha}{\alpha - 1}\right)^2$$

We will again leverage weak duality of semidefinite programs to upper-bound the objective function of 7, and thus achieve an upper-bound for 2. In order to do so, we first introduce dual variables for each of the constraints of 7:

$$\lambda_1 \geq 0, \quad \lambda_2 \geq 0, \quad \lambda_3 \geq 0, \quad \lambda_4 \geq 0, \quad \lambda_5 \geq 0$$

As is standard practice in deriving a dual semidefinite program, we first construct the following constraint (for all values of $q, v, a, t$):

$$\mathcal{L}(t, q, v, a) = \lambda_1 \left(a^T a - 2a^T t + t^T t - \frac{v^T v - 2v^T a + a^T a}{\alpha^2}\right) - \lambda_2 \left(a^T a - 2a^T q + q^T q\right)$$
$$+ \lambda_3 \left(a^T a - 2a^T q + q^T q\right) + \lambda_4 \left(v^T v - 2v^T t + t^T t - v^T v + 2v^T a - a^T a\right)$$
$$+ \lambda_5 \left(q^T q - 2q^T v + v^T v\right) - \left(t^T t - 2t^T q + q^T q\right)$$

This gives the following dual program:

$$\min \lambda_2 - \lambda_3 + \lambda_5 \left(\beta + \frac{\alpha}{\alpha - 1}\right)^2 \quad \text{such that:} \tag{9}$$
$$\mathcal{L}(t, q, v, a) \geq 0$$

Now, consider the following values for each of the dual variables:

$$\lambda_1 = \frac{1}{\alpha^2} + \left(1 - \frac{1}{\alpha}\right)^2 \frac{\beta(\alpha^2 - 1) + \alpha^2}{\alpha(\alpha - 1)^2 \beta^2 + (\alpha^2 - 1)(2\alpha - 1)\beta + (2\alpha - 1)\alpha^2}$$

$$\lambda_2 - \lambda_3 = \left(\frac{\beta}{\alpha} + \frac{\alpha}{\alpha - 1}\right)^2 - \frac{1}{\alpha^2}\left(\beta + \frac{\alpha}{\alpha - 1}\right)\left(\beta + \frac{\alpha^2}{\alpha - 1}\right)$$

$$\lambda_4 = \frac{\beta(\alpha - 1)^2 + \alpha^2(\alpha - 1)}{\alpha(\alpha - 1)^2 \beta^2 + (\alpha^2 - 1)(2\alpha - 1)\beta + (2\alpha - 1)\alpha^2}, \quad \lambda_5 = \frac{1}{\alpha^2}\frac{\beta + \frac{\alpha^2}{\alpha - 1}}{\beta + \frac{\alpha}{\alpha - 1}}$$

Under these dual variables, $\mathcal{L}(t, q, v, a)$ simplifies to a perfect square, thus implying non-negativity of the values $t, q, v$, and $a$, and the feasibility of the dual constraints. Therefore $\left(\frac{\beta}{\alpha} + \frac{\alpha}{\alpha - 1}\right)^2$ is the value of the dual optimization function, giving an upper-bound on the optimal value from equation 7 by weak duality.

$$\square$$

### B.3. Proof for Theorem 1.1

With the above lemmas we are now ready to prove Theorem 1.1 and we begin by stating a few intermediate results.

Let $v$ be the vertex whose out-neighborhood $N_{\text{out}}(v)$ is explored in the $i^{\text{th}}$ step. A step is *successful* if the first point of change in the beam is above (or before) the position of $v$ in the beam: $D(v, q) > \min_{x \in (N_{\text{out}}(v) \setminus B^{(i)})} D(x, q))$, where $B^{(i)}$ denotes the state of the beam at step $i$. Otherwise, we call the step *unsuccessful*. Intuitively, an unsuccessful step makes no progress towards convergence, as no expanded out-neighbor has been added that is closer than $v$ to $q$.

First, we show that after $i$ unsuccessful steps, the $i$th member of the beam, denoted by $v_i$, is arbitrarily close to the $i^{\text{th}}$ member of the locally optimal beam denoted by $a_i$.

**Lemma B.1.** *After $i$ unsuccessful steps, the following holds for $i^{th}$ member $v_i$ of the beam:*

$$D(v_i, q) \leq \alpha_{\text{opt}} \cdot D(a_i, q)$$

*Proof.* We call a node $v$ in the beam *inactive* if its neighborhood is explored *and* it results in an unsuccessful step. If a node $v_p$ at position $p$ in the beam is *inactive*, then the first $p$ members of the beam constitute a locally optimal beam, and by Lemma 3.6 we have:

$$D(v_p, q) \leq \alpha_{\text{opt}} \cdot D(a_p, q). \tag{10}$$

Note that a node can become freshly *inactive* only if all the nodes above it are *inactive*. After $i$ unsuccessful steps, there are $i$ inactive nodes, and each of these nodes became inactive at a position less than or equal to index $i$. Therefore, by Equation 10, each of these $i$ inactive nodes has distance to $q$ at most $\alpha_{\text{opt}} \cdot D(a_i, q)$.

Since the $i$-th element in the current beam, denoted by $v_i$, is either one of these inactive nodes or has distance to $q$ less than or equal to that of at least one of these $i$ inactive nodes, it follows that:

$$D(v_i, q) \leq \alpha_{\text{opt}} \cdot D(a_i, q).$$

This completes the proof.

$\square$

Next, we show that after a series of successful steps, the $i^{\text{th}}$ node in the beam upon completion of those steps gets close to the query as summarized in the lemma below.

**Lemma B.2.** *Consider a DiskANN graph $G$ on $n$ points with respect to a metric $D$, such that the maximal value of optimization problem 1 under $D$ is $\alpha_{\text{opt}}$ and the optimal value of optimization problem 2 is less than $\frac{\beta}{\alpha}$. Consider a query $q$ and let the $i^{\text{th}}$ member of the beam after $S$ successful steps be $v_{(i,S)}$. The following then holds for $v_{(i,S)}$:*

$$D(v_{(i,S)}, q) \leq \frac{\Gamma}{\alpha^{S-i}} + \max\left(\alpha_{\text{opt}}, \frac{\alpha^2 + 1}{\alpha^2 - 1}\right) \cdot D(a_i, q)$$

*Proof.* We proceed with proof by induction over the number of successful steps S.

**Base case:** For $S < i$ steps, see that the following holds by the triangle inequality:

$$D(v_{(i,S)}, q) \leq D(v_{(i,S)}, a_i) + D(a_i, q) \leq \Gamma + D(a_i, q)$$

**Induction claim**: For all $S < j$, we claim that the following holds :

$$D(v_{(i,S)}, q) \leq \frac{\Gamma}{\alpha^{S-i}} + \max\left(\alpha_{\text{opt}}, \frac{\alpha^2 + 1}{\alpha^2 - 1}\right) \cdot D(a_i, q)$$

**Induction Step**: Let $LI(k)$ be the lowest index in the beam that changes in the $k^{\text{th}}$ successful step. Then, $v_{LI(k)}$ must be the closest node to $q$ introduced by $k^{th}$ successful step. Note that as a node's index in the beam gets closer to 1, its proximity to $q$ also decreases.

If $LI(j) > i$, then the beam given by $\left[v_{(1,j-1)}, ...v_{(i,j-1)}\right]$ is a locally optimal by the definition of a successful step and of $LI(j)$. Therefore, if $LI(j) > i$, Lemma 3.6 gives us:

$$D(v_{(i,j)}, q) \leq \alpha_{\text{opt}} \cdot D(a_i, q)$$

As a result, we need only show that for all steps $k < j$, the following holds: $LI(k) \leq i$.

If $LI(j) < i$, then $D(v_{(i,j)}, q)$ must be upper-bounded by $D(v_{(i-1,j-1)}, q)$, as $v_{(i-1,j-1)}$ would be pushed back in the beam by $v_{(j,LI(j))}$. Therefore,

$$D(v_{(i,j)}, q) \leq D(v_{(i-1,j-1)}, q)$$
$$\leq \frac{\Gamma}{\alpha^{(j-1)-(i-1)}} + \alpha_{\text{opt}} \cdot D(a_{i-1}, q)$$

If $LI(j) = i$, we run into two cases. For the case where the neighborhood of some node below $v_{(i,j-1)}$ is explored, $\left[v_{(1,j-1)}, ...v_{(i,j-1)}\right]$ must be a locally optimal beam. Therefore, we get:

$$D(v_{(i,j)}, q) \leq \alpha_{\text{opt}} \cdot D(a_i, q)$$

If $v_{(i,j-1)}$'s neighborhood is explored we have two (sub-)cases. For the case where $v_{(i,j)} = a_i$, the approximation ratio trivially holds. Next, consider the case where $V_{(i,j)} \neq a_i$. Since the slow-preprocessing version of DiskANN is sorted $\alpha$-reachable, we can define the following function for any pair $(p, q)$:

$$f(p, q) = \begin{cases} q, & \text{if (p,q)} \in E \\ \underset{i \in N_{\text{out}}(p), D(p,i) \leq D(p,q)}{\arg\min} D(i, q) & \text{otherwise} \end{cases}$$

Let $a_o$ be the closest vertex to $q$ which is not in the beam. Consider a series of nodes $S$ where

$$S_t = \begin{cases} v_{(i,j-1)}, & \text{if } t = 1 \\ f(S_{t-1}, a_o), & \text{otherwise} \end{cases}$$

See that $S_{\text{inf}} = a_o$, and by the definition of sorted $\alpha$-reachability, we have

$$D(S_{t+1}, a_o) \leq \frac{D(S_t, a_o)}{\alpha}$$

Now, let $t$ be the smallest index such that $S_t$'s neighborhood has not been explored by the beam search. Note that such an index always exists because $S_1 = v_{(i,j-1)} \neq a_o$ and $S_{\text{inf}} \notin B$, where B denotes the beam)

After $j - 1$ steps, see that

$$D(v_{(i,j-1)}, q) = \beta + \max\left(\alpha_{\text{opt}}, \frac{\alpha^2 + 1}{\alpha^2 - 1}\right) \cdot D(a, q)$$

In addition, we can show after $j$ steps

$$D(v_{(i,j)}, q) \leq \frac{\beta}{\alpha} + \max\left(\alpha_{\text{opt}}, \frac{\alpha^2 + 1}{\alpha^2 - 1}\right) \cdot D(a, q) \tag{11}$$

We can now examine If $t \geq 3$,

$$
\begin{aligned}
D(S_t, q) &\leq D(S_t, a_o) + D(a_o, q) \quad \text{(triangle inequality)} \\
&\leq \frac{D(v_{(i,j-1)}, a_o)}{\alpha^2} + D(a_o, q) \quad \text{(definition of } S_t\text{)} \\
&\leq \frac{D(v_{(i,j-1)}, q) + D(q, a_o)}{\alpha^2} + D(a_o, q) \quad \text{(triangle inequality)} \\
&\leq \frac{D(v_{(i,j-1)}, q)}{\alpha^2} + \frac{\alpha^2 + 1}{\alpha^2} D(a_o, q) \quad \text{(definitions of } S_t\text{)} \\
&\leq \frac{\beta + \max\left(\alpha_{\text{opt}}, \frac{\alpha^2+1}{\alpha^2-1}\right) \cdot D(a_o, q)}{\alpha^2} + \frac{\alpha^2 + 1}{\alpha^2} D(a_o, q) \quad \text{(Equation (11))} \\
&\leq \frac{\beta}{\alpha^2} + \frac{\alpha^2 + 1 + \max\left(\alpha_{\text{opt}}, \frac{\alpha^2+1}{\alpha^2-1}\right)}{\alpha^2} D(a_o, q) \quad \text{(definitions of } S_t\text{)} \\
&\leq \frac{\beta}{\alpha^2} + \max\left(\alpha_{\text{opt}}, \frac{\alpha^2 + 1}{\alpha^2 - 1}\right) D(a_o, q)
\end{aligned}
$$

For $t = 2$, we know $S_2$ is an out-neighbor of $v$. Therefore, we can write

$$
D(a, t) \leq \frac{D(v, a)}{\alpha}
$$

$$
D(v_{(i,j-1)}, S_t) \leq D(v, a)
$$

We can also rescale all the distances to make $D(a, q)$ to be unit length, giving us

$$
D(a, q) = 1
$$

$$
D(q, v) \leq \beta' + \alpha_{\text{opt}}
$$

where $\beta'$ is $\beta$ after rescaling. In addition, by Equation (2), the distance between $S_t$ and $q$ is bounded by $\alpha_{\text{opt}} + \frac{\beta'}{\alpha}$.
Therefore, we get

$$
D(v_{(i,j)}, q) \leq \frac{\beta}{\alpha} + \max\left(\alpha_{\text{opt}}, \frac{\alpha^2 + 1}{\alpha^2 - 1}\right) D(a, q)
$$

and by comparing with equation from induction step

$$
D(v_{(i,j)}, q) \leq \frac{\Gamma}{\alpha^{j-i}} + \max\left(\alpha_{\text{opt}}, \frac{\alpha^2 + 1}{\alpha^2 - 1}\right) D(a, q)
$$

$\square$

**Theorem 1.1** (Final convergence bounds). *Let $G$ be a DiskANN graph. For any query $q$,* BeamSearch *of size $L$ in at most $O(L + \log_\alpha \frac{\delta}{(\alpha-1)\epsilon})$ steps outputs a set of points $\{b_1, \ldots, b_L\}$ such that each $b_j$ satisfies:*

$$
D(b_j, q) \leq \epsilon + \frac{\alpha}{\alpha - 1} \cdot D(a_j, q) \text{ for } \ell_2 \text{ metric } D
$$

$$
D(b_j, q) \leq \epsilon + \frac{\alpha + 1}{\alpha - 1} \cdot D(a_j, q) \text{ for any metric } D
$$

*where $a_j$ is the $j$th nearest neighbor to $q$.*

*Proof.* Combining Lemma B.1 and Lemma B.2 we get that after $i$ steps the following inequality holds,

$$
D(v_{(l,i)}, q) \leq \frac{\Gamma}{\alpha^{i-l}} + \max\left(\alpha_{\text{opt}}, \frac{\alpha^2 + 1}{\alpha^2 - 1}\right) D(a, q)
$$

which can be seen as parallel from inequality (1) from Theorem 3.4 in (Indyk & Xu, 2024). Further substituting the value of $\alpha_{\mathrm{opt}}$ from Lemma 3.8 for general metric and from Lemma 3.9 for Euclidean metric and following the proof similar to that of (Indyk & Xu, 2024) we get our main result. □

