# OpenReview forum: "Sort Before You Prune: Improved Worst-Case Guarantees of the DiskANN Family of Graphs"
_ICML.cc/2025/Conference — ICML 2025 poster_

### Official Review · Reviewer_eMvk · 2025-03-11

**Overall Recommendation:** 4

**Summary:**

This paper aims to strengthen the theoretical analysis of graph-based ANN algorithms. In DiskANN indexing, candidate vertices are considered in ascending order from their distance to the out-vertex $v$, a detail not leveraged in prior theoretical analysis works. By combining this sorted-distance implementation detail with the observation that most graph ANN indices are built over the Euclidean metric, rather than an arbitrary one, the work improves the distance approximation factor that such a graph index is guaranteed to satisfy. The work then analyzes beam search rather than greedy top-1 search and shows an analogous result.

**Claims And Evidence:**

The claims are well-supported.

**Essential References Not Discussed:**

The essential references, to the best of my knowledge, are discussed.

**Experimental Designs Or Analyses:**

The experiments make sense.

**Methods And Evaluation Criteria:**

The experiments make sense.

**Other Comments Or Suggestions:**

N/A

**Other Strengths And Weaknesses:**

This paper is very well written and easy to follow, and its extension of upper bound analysis to beam search is significant, because beam search is always used in practice in graph ANN algorithms. One weakness is that the work gives a distance approximation bound, which is difficult for typical ANN users to interpret. A probabilistic analysis of finding the exact top-1 neighbor would be more relevant, although difficult. The degree and step bounds are also quite weak still and often lead to trivial (brute-force level) upper bounds.

**Questions For Authors:**

N/A

**Relation To Broader Scientific Literature:**

This work improves the theoretical analysis of graph-based ANN algorithms. It is very much related to the Indyk & Xu paper it references extensively, and also focuses on a variant of the DiskANN indexing procedure.

**Theoretical Claims:**

The proofs for the two main results both appear correct to me.

---

> ### Author Rebuttal · Authors · 2025-04-01
>
> We thank the reviewer for the kind words and suggestions.  As the reviewer noted, extending the existing analysis to the practically motivated beam search regime was a key motivation for this work. Additionally, we adopted a distance-based approximation scheme, leveraging our novel factor-revealing SDP to establish improved approximation guarantees for the Euclidean metric. While we agree that a probabilistic analysis of graph-based data structures would be valuable, it is important to note that DiskANN is deterministic and always returns the same set of nearest neighbors for a given query. Therefore, conducting a probabilistic analysis of exact top-1 nearest neighbor retrieval would require a new theoretical framework with a well-motivated assumption on the query distribution—an interesting yet challenging direction for future research.

---

### Official Review · Reviewer_fRUc · 2025-03-13

**Overall Recommendation:** 4

**Summary:**

This paper introduces a novel strategy for constructing graph structures to enhance Approximate Nearest Neighbor Search (ANNS) on high-dimensional vectors. This approach ensures a superior approximation ratio for L2 distance metrics compared to existing methods.

## **Update After Rebuttal**
The rebuttal has carefully addressed these comments. Since I have no more concerns now, I have raised my score.

**Claims And Evidence:**

Regarding the theoretical results, the claims are generally okay for me. However, the evaluations have certain issues (please refer to the following comments).

**Essential References Not Discussed:**

N/A

**Experimental Designs Or Analyses:**

In terms of the experimental designs, I have the following comments:

1. **Dataset Scalability**: The datasets used in the experiments are relatively small. The authors should evaluate their method on larger-scale datasets containing over 100 million or even 1 billion vectors to demonstrate scalability and robustness. In particulr, the competitor DiskANN is designed for in-disk index, which aims to handle billion-scale vectors.

2. **Evaluation Scope**: The experimental study focuses only on the application scenarios of in-memory indexes. Except for the new graph structure, DiskANN also contributes to ANNS over large-scale data that needs to be stored on disk. This application scope should be emphasized and validated in the experiments."

3. **Baseline Selection**: If authors try to claim that their contributions lay on the novel graph structure, they may have to compare their methods with other existing graph-based ANNS index like NSG [PVLDB2019] and NSSG [TPAMI22].

[VLDB2019] Cong Fu, Chao Xiang, Changxu Wang, et al. Fast Approximate Nearest Neighbor Search With The Navigating Spreading-out Graph. PVLDB, 12(5): 461-474, 2019.

[TPAMI22] Cong Fu, Changxu Wang, Deng Cai. High Dimensional Similarity Search With Satellite System Graph: Efficiency, Scalability, and Unindexed Query Compatibility. IEEE Trans. Pattern Anal. Mach. Intell. 44(8): 4139-4150 (2022)

=====================
The rebuttal has carefully addressed these comments, so I have no concerns on these issues now.

**Methods And Evaluation Criteria:**

The proposed sorted α-reachable graphs can achieve faster search performances than the existing method DiskANN. However, in the experimental studies, this paper only conducts evaluations on 3 relatively small-scale datasets, and only reports the results of the search efficiency. Quite a few critical metrics are ignored, so the proposed solution may not be practical enough under the general evaluation criterias under benchmarks of ANNS.

**Other Comments Or Suggestions:**

N/A

**Other Strengths And Weaknesses:**

N/A

**Questions For Authors:**

1. How does varying k (e.g., increasing from 20 to 100) affect search performance?

2. For OpenAI dataset (dimension = 1536), the average node degree decreases from 56 to 53, while for SIFT1M (dimension = 128), it decreases from 49 to 44 (a similar trend is observed in the Wiki dataset). Does this imply that the performance of the proposed strategy diminishes as the dimensionality of the vectors increases?

**Relation To Broader Scientific Literature:**

This paper propose a new strategy for ANNS index with a better approximation ratio.

**Theoretical Claims:**

The theoretical results look okay and may offer insights to existing research. However, the proposed approximation ratio appears to be limited under L2 distance.

---

> ### Author Rebuttal · Authors · 2025-04-01
>
> We appreciate the reviewer’s detailed feedback and would like to clarify the key contributions of our work. Rather than proposing a new algorithm that outperforms DiskANN, our primary contribution is theoretical, providing a deeper understanding of the DiskANN algorithm in two key ways.
> First, we establish an improved approximation guarantee for DiskANN by leveraging two critical insights: (a) explicitly analyzing the $\ell_2$ setting, which is widely used in practice and has favorable mathematical properties, and (b) examining the _sorting_ step in the prune procedure. Using these insights, we introduce a novel factor-revealing SDP to demonstrate an improved approximation factor of $\frac{\alpha}{(\alpha - 1)}$, compared to the previously known bound of $\frac{\alpha + 1}{\alpha - 1}$ from Indyk and Xu (NeurIPS 2023). Moreover, we show that omitting either sorting or the $\ell_2$ setting results in a best possible approximation ratio of $\frac{\alpha + 1}{\alpha - 1}$, matching Indyk-Xu’s original result. This suggests that DiskANN may perform better under the $\ell_2$ metric than other distance metrics.
>
> Second, we introduce techniques that establish the first-ever theoretical bounds for retrieving $k > 1$ candidates in a beam search setting using a graph-based ANNS algorithm. Despite its practical significance in scalable retrieval, this regime remains underexplored, with prior work (NeurIPS 2023, ICML 2020, SoCG 2018, SODA 1993) exclusively focusing on the $k = 1$ case. We also believe our factor-revealing LP/SDP framework will be valuable for further research in this setting. In short, our core contribution lies in the theoretical understanding of the sorted prune step in DiskANN, and to support our theoretical findings, we conducted experiments evaluating the impact of sorting during graph construction. We did not compare DiskANN with other ANNS algorithms, as this has been extensively studied in prior work.
>
> We thank the reviewer for suggesting additional experiments. Based on the suggestions, we include plots and tables for 100M-scale datasets, key metrics such as build times, and RAM usage and disk I/O reads per query (for SSD-based indices). Please note that all dropbox links are anonymized. This comprehensive empirical study will be included in the final version.
>
> Link to DiskIO plots: https://www.dropbox.com/scl/fo/c44n8ij92zpt19qdedw1o/AJ-GeHiB6-38q1eRqJ7XYSw?rlkey=yfvqwr98038o4xuvq9dj1ghcy&st=ws3cd67g&dl=0
>
> Link to RAM usage and Build Times: https://www.dropbox.com/scl/fo/v3cdv5iv76aryqcrh639x/AJgfthBn3IFAVaUGdWGzHwU?rlkey=ghddj7evgo7ldcrl1whc6st7m&st=nsjbz81n&dl=0
>
> Link to Large-Scale dataset plots: https://www.dropbox.com/scl/fo/mp2cq55hz6bbd7rswojyu/AHJF6oHGCVUt5xz6Ah-OPao?rlkey=rt7dvticzt2jlsvq3pkf4nncf&st=j4saygfe&dl=0
>
> **Q1 (varying k)**:
>
> On two large scale datasets, we have included plots of Recall@k vs. QPS for different values of k to analyze the effect of sorting during pruning. The search behavior remains consistent with our observations for k= 100.
>
> Link to varied-k plots: https://www.dropbox.com/scl/fo/m6f1pdq01npiy0dlv2ueq/AJa2AL5Ij3J1S8hhkvVhax8?rlkey=6f83rhwgkafme6s6v6r4ke3v6&st=um503h2h&dl=0
>
> **Q2 (degree reduction and varying dimension)**:
>
> We appreciate the reviewer’s insightful question. While the degree does decrease for graphs built using sorted pruning on real-world datasets (as shown in Table 2), there is no clear correlation between this decrease and data dimensionality.
> Consider a simple one-dimensional example: let $p, a, b, c$ be four points on a line with distances $d(p,a)=1$, $d(a,b)=0.5$, and $d(b,c)=1.5$. The sorted algorithm would connect p to both $a$ and $c$ to maintain $\alpha$-reachability, whereas a more efficient algorithm could instead connect $p$ to $b$ while still preserving $\alpha$-reachability. This example illustrates that degree change is not tied to dimensionality.
> Additionally, a lower degree does not necessarily translate to improved performance, as latency depends on both the average degree and the graph’s diameter. A reduced degree may increase the graph’s diameter, making the impact on final latency difficult to predict.
> To further investigate the effect of data dimension on the performance of sorted vs unsorted index construction, we conducted an additional experiment on the OpenAI dataset. We used a random Johnson-Lindenstrauss matrix to project the 1536-dimensional OpenAI embeddings into a 352-dimensional space and measured the degree change due to sorting. With sorting, the average degree was 54.15; without sorting, it increased to 57.3—a 5.8% rise. This aligns closely with the 5.6% increase observed in the original dataset (Table 2), suggesting that the degree change is a property of the dataset rather than a direct function of dimensionality. So it might be more a function of the inherent structure of the dataset as opposed to the dimension itself.

---

> > ### Comment · Reviewer_fRUc · 2025-04-02
> >
> > In the rebuttal, the authors have carefully addressed my previous comments, particularly regarding the experimental design and analysis. After reviewing the updated results and discussion, I have no further concerns at this time. Therefore, I will raise my score based on the rebuttal.

---

### Official Review · Reviewer_gnMJ · 2025-03-14

**Overall Recommendation:** 3

**Summary:**

This paper conducts a theoretical analysis of graph-based approximate nearest neighbor search algorithms. In particular, the authors propose a new theoretical framework called the $\alpha$-reachable graph and, using this framework, provide the first worst-case complexity analysis of beam search in situations where $k > 1$.

## update after rebuttal
After the rebuttal, I decided to keep my original score, WA.

**Claims And Evidence:**

Yes.

**Essential References Not Discussed:**

None.

**Ethical Review Concerns:**

None.

**Experimental Designs Or Analyses:**

Yes.

**Methods And Evaluation Criteria:**

Yes.

**Other Comments Or Suggestions:**

In Sec A.4 of the supplementary material, the statement "hence we can reformulate 6 as 1" appears to be a typo and should likely be "hence we can reformulate objective 6 as problem 1." Similarly, there is inconsistency throughout the paper regarding whether to write "1" or "problem 1." I recommend unifying this notation. If only a number is used, it would be more precise to enclose it in parentheses as "(1)."

**Other Strengths And Weaknesses:**

## Strengths

The greatest strength of this paper lies in its extension of [Indyk & Xu, 2024] to analyze the worst-case complexity of beam search for $k > 1$.

Introducing the $\alpha$-reachable concept is intuitive, and the problem setup is straightforward without conceptual difficulties.

Additionally, Problem 1 in Sec 3.1 is formulated clearly, and the restructuring of the problem in this way is intriguing. The proofs are elementary to follow.

Moreover, the insight that a dual problem can be considered when transitioning from a general metric $D$ to the Euclidean distance is well-motivated and interesting.

## Weaknesses

There are no major concerns, but if anything, the experiments in Sec 4 focus on illustrating the concept of "sorting" rather than supporting the paper's primary claim regarding worst-case complexity. It would be interesting to see a more direct evaluation of how the proposed bounds relate to general high-dimensional vector data.

**Questions For Authors:**

I have no particular questions.

**Relation To Broader Scientific Literature:**

This paper advances the theoretical development of graph-based algorithms in approximate nearest neighbor search. In particular, it is a direct extension of [Indyk & Xu 2024].

**Theoretical Claims:**

Yes.

---

> ### Author Rebuttal · Authors · 2025-04-01
>
> We thank the reviewer for their thoughtful comments and suggestions. We will address all typos and wording mistakes in the final version. We want to emphasize one point: in addition to giving the first analysis for beam search, which is widely used in practice, we also improve the current state of the art approximation guarantees for the DiskANN algorithm, when the underlying metric is euclidean. This suggests that DiskANN may perform better under the $\ell_2$ metric compared to other distance metrics.
>
> Regarding experiments for quantifying the solution quality, we include a link to plots demonstrating average approximation ratio across queries versus QPS for two large 100M-scale datasets: Microsoft SPACEV and BIGANN (https://big-ann-benchmarks.com/neurips23.html) for different values of $k$. The average approximation ratio is the mean over all queries, of the maximum over all items $i$ in the final beam of size $k$, of the distance of the $i$th candidate computed by the algorithm upon the $i$th closest NN.
>
> (Anonymized Dropbox link) Approximation Ratio Plots: https://www.dropbox.com/scl/fo/y76d8hn51pkl097g6yaa0/AGyI89kxLaI8ouvgG8FNjRM?rlkey=xahf5obtefjqz35jj1yig6p3r&st=5h6r0brf&dl=0
>
> For convenience, we include the raw data for the BIGANN approximation ratio plot below. Note that for an approximation ratio of 1.008, the sorted instance supports a QPS of around 58000 (row 7)  while the unsorted instance supports a QPS of only 25000 (row 11).
>
> | Row | Unsorted Instance QPS | Sorted Instance QPS | Unsorted Approximation Ratio | Sorted Approximation Ratio |
> |-------|-----------------------|---------------------|-----------------------------|---------------------------|
> | 1     | 84464.5              | 93165.4            | 1.02061                     | 1.01429                   |
> | 2     | 78923.5              | 85955.7            | 1.01918                     | 1.01336                   |
> | 3     | 76587.9              | 79858.2            | 1.01795                     | 1.01257                   |
> | 4     | 71633.8              | 76934.9            | 1.01704                     | 1.01140                   |
> | 5     | 67506.9              | 67532.4            | 1.01612                     | 1.01064                   |
> | 6     | 59775.7              | 61504.0            | 1.01384                     | 1.00933                   |
> | 7     | 53310.9              | 58497.7            | 1.01267                     | 1.00846                   |
> | 8     | 51330.2              | 54442.5            | 1.01221                     | 1.00796                   |
> | 9     | 40633.4              | 42155.7            | 1.01029                     | 1.00675                   |
> | 10    | 32898.3              | 32861.8            | 1.00907                     | 1.00579                   |
> | 11    | 27124.3              | 28039.5            | 1.00812                     | 1.00524                   |
> | 12    | 22319.7              | 23551.2            | 1.00726                     | 1.00451                   |

---

### Official Review · Reviewer_USgk · 2025-03-14

**Overall Recommendation:** 4

**Summary:**

This paper addresses the problem of Approximate Nearest Neighbor Search (ANNS), which is crucial for various applications dealing with large datasets in high-dimensional spaces. Graph-based data structures, particularly those in the DiskANN family, have shown strong empirical performance. However, theoretical understanding of their worst-case behavior has been limited. This work builds upon the concept of alpha-reachable graphs introduced by for DiskANN, which provided the first provable worst-case guarantees. The authors identify three key open questions from that prior work:
1.  Can stronger guarantees be derived for the Euclidean  metric, commonly used in practice?
2.  Does the sorting step before pruning in the DiskANN algorithm contribute to the theoretical guarantees?
3.  Can provable guarantees be established for beam search, the practical algorithm used to retrieve k > 1 nearest neighbors?
This paper aims to answer these questions, offering improved theoretical analysis for DiskANN, especially in Euclidean spaces, and providing the first worst-case guarantees for beam search in graph-based ANNS methods for retrieving multiple neighbors.

**Claims And Evidence:**

Authors introduce sorted α-reachable graphs, and use this notion to obtain a stronger approximation factor of α/(α−1) for the DiskANN algorithm on Euclidean metrics. Authors present the first worst-case theoretical analysis for the popular beam-search algorithm, which is used in practice to search these graphs for k > 1 candidate nearest neighbors.

**Essential References Not Discussed:**

-

**Experimental Designs Or Analyses:**

Authors provided benchmarks on three datasets: OpenAI, SIFT-1M, and Wikipedia, with the baseline DiskANN implementation from ParlayANN. The benchmark and analysis compared the graph degrees, recall, and QPS on modern hardwares.

It could be interesting to see if the method scale to larger datasets that big-ann provides (https://big-ann-benchmarks.com/neurips23.html). Or if the method generalizes to other small datasets provided by ann-benchmarks (https://ann-benchmarks.com/index.html)

**Methods And Evaluation Criteria:**

The proposed algorithm is similar to the DiskAnn graph, except that the proposed method e sort the candidate neighbors before pruning the graph, and the baseline method does not. Authors then compare the graph degrees, recall, and QPS on modern hardwares.

**Other Comments Or Suggestions:**

-

**Other Strengths And Weaknesses:**

-

**Questions For Authors:**

Can authors provide more benchmarks such as the big-ann (https://big-ann-benchmarks.com/neurips23.html) or  ann-benchmarks (https://ann-benchmarks.com/index.html)?

**Relation To Broader Scientific Literature:**

-

**Theoretical Claims:**

Authors define the sorted alpha reachable graph as follows. Given a dataset P, a distance metric D and α > 1, a directed graph G to be a sorted α-reachable graph if for any pair of points v, a ∈ P, either the edge (v, a) exists in G or there exists a point t ∈ P such that:
1. (v, t) is an edge in G
2. D(t, a) ≤ D(v, a)/α (α-reachability)
3. D(v, t) ≤ D(v, a) (sorting property)
Author argues that when performing beam search on a sorted alpha reachable graph, the worst case is tight on the bounds:
D(b_j , q) ≤ α/(α − 1) · D(a_j , q) for L2 metric, and D(b_j , q) ≤ (α + 1)/(α − 1) · D(a_j , q) for general metric, where a_j is the j'th nearest neighbor to q.

---

> ### Author Rebuttal · Authors · 2025-04-01
>
> We thank the reviewer for their kind words and thoughtful suggestions. As suggested by the reviewer, we have included a link to plots on two large 100M-scale datasets from big-ann-benchmark, confirming the importance of sorting irrespective of dataset size. We will include all of these plots in the final version of our submission. For convenience, we also include a table capturing one such plot for the 100M-scale Microsoft SPACEV dataset, comparing QPS to 100@100Recall. See that for this dataset, the sorted index outperforms the unsorted index, delivering 23% higher QPS for a recall of ~0.93.
>
> (Anonymized Dropbox link) Large Scale Experiment Plots: https://www.dropbox.com/scl/fo/mp2cq55hz6bbd7rswojyu/AHJF6oHGCVUt5xz6Ah-OPao?rlkey=rt7dvticzt2jlsvq3pkf4nncf&st=7z86irq4&dl=0
>
> | Row | Unsorted QPS | Unsorted 100@100 Recall | Sorted QPS | Sorted 100@100 Recall |
> |-------|---------------------|-------------------------------|-----------------------|----------------------------------|
> | 1     | 98,408              | 0.9044                        | 108,715               | 0.9132                           |
> | 2     | 88,007              | 0.9101                        | 95,070                | 0.9184                           |
> | 3     | 80,488              | 0.9153                        | 88,984                | 0.9228                           |
> | 4     | 76,305              | 0.9197                        | 84,397                | 0.9268                           |
> | 5     | 71,054              | 0.9235                        | 78,176                | 0.9301                           |
> | 6     | 63,347              | 0.9302                        | 68,483                | 0.9358                           |
> | 7     | 57,392              | 0.9357                        | 61,069                | 0.9405                           |
> | 8     | 54,370              | 0.9381                        | 58,041                | 0.9425                           |
> | 9     | 42,049              | 0.9479                        | 44,921                | 0.9505                           |
> | 10    | 33,312              | 0.9541                        | 35,845                | 0.9560                           |
> | 11    | 27,583              | 0.9587                        | 29,182                | 0.9602                           |
> | 12    | 22,502              | 0.9621                        | 23,804                | 0.9634                           |

---

### Decision · Program_Chairs · 2025-05-01

**Decision:**

Accept (poster)

**Comment:**

The paper provides an innovation for, a theoretical understanding of and an approximation worst-case guarantee for a widely used approximate nearest neighbor search known as DiskANN. The proposed additional sorting step is shown to improve performance and for which worst-case performance is provided.  Empirical results support both the advantages of the sorting step as well as its performance particularly on one dataset.